# Investigating the Relationship between Vitamin D and Persistent Symptoms Following SARS-CoV-2 Infection

**DOI:** 10.3390/nu13072430

**Published:** 2021-07-15

**Authors:** Liam Townsend, Adam H. Dyer, Patrick McCluskey, Kate O’Brien, Joanne Dowds, Eamon Laird, Ciaran Bannan, Nollaig M. Bourke, Cliona Ní Cheallaigh, Declan G. Byrne, Rose Anne Kenny

**Affiliations:** 1Department of Infectious Diseases, St James’s Hospital, Dublin 8, Ireland; cbannan@stjames.ie (C.B.); nicheacm@tcd.ie (C.N.C.); 2Department of Clinical Medicine, School of Medicine, Trinity Translational Medicine Institute, Trinity College, Dublin 2, Ireland; declangbyrne@gmail.com; 3Department of Medical Gerontology, School of Medicine, Trinity Translational Medicine Institute, Trinity College, Dublin 8, Ireland; dyera@tcd.ie (A.H.D.); lairdea@tcd.ie (E.L.); NBOURKE@tcd.ie (N.M.B.); 4Department of Clinical Medicine, St James’s Hospital, Dublin 8, Ireland; mccluskp@tcd.ie; 5Department of Physiotherapy, St James’s Hospital, Dublin 8, Ireland; kateobrien@stjames.ie (K.O.); jdowds@stjames.ie (J.D.); 6The Irish Longitudinal Study on Ageing (TILDA), Trinity College Dublin, Dublin 2, Ireland; rkenny@tcd.ie; 7Falls and Syncope Unit, Mercer’s Institute for Successful Ageing, St James’s Hospital, Dublin 8, Ireland

**Keywords:** vitamin D, long COVID, fatigue, SARS-CoV-2

## Abstract

The emergence of persistent symptoms following SARS-CoV-2 infection, known as *long COVID*, is providing a new challenge to healthcare systems. The cardinal features are fatigue and reduced exercise tolerance. Vitamin D is known to have pleotropic effects far beyond bone health and is associated with immune modulation and autoimmunity. We hypothesize that vitamin D levels are associated with persistent symptoms following COVID-19. Herein, we investigate the relationship between vitamin D and fatigue and reduced exercise tolerance, assessed by the Chalder Fatigue Score, six-minute walk test and modified Borg scale. Multivariable linear and logistic regression models were used to evaluate the relationships. A total of 149 patients were recruited at a median of 79 days after COVID-19 illness. The median vitamin D level was 62 nmol/L, with *n* = 36 (24%) having levels 30–49 nmol/L and *n* = 14 (9%) with levels <30 nmol/L. Fatigue was common, with *n* = 86 (58%) meeting the case definition. The median Borg score was 3, while the median distance covered for the walk test was 450 m. No relationship between vitamin D and the measures of ongoing ill-health assessed in the study was found following multivariable regression analysis. These results suggest that persistent fatigue and reduced exercise tolerance following COVID-19 are independent of vitamin D.

## 1. Introduction

SARS-CoV-2, the causative virus of the current COVID-19 pandemic, produces a wide spectrum of disease. Initial infection can range from mild symptoms, such as loss of taste and anosmia, to severe presentations characterized by marked inflammation, coagulopathy and respiratory failure, sometimes leading to death [1,2,3]. Endothelial dysfunction is also a hallmark of acute SARS-CoV-2 infection [4,5]. Similarly, the convalescent period following infection is characterized by wide symptom variability. Persistent symptoms following COVID-19, known as post-acute COVID-19 syndrome (PACS) or *long COVID*, are common [6]. Given that up to one-third of patients have persistent symptoms up to six months after infection, there is a pressing need to identify the underlying pathological processes [7]. The cardinal features of *long COVID* are broad, including cognitive issues and cough, but the cardinal features are fatigue and reduced exercise tolerance [8,9]. However, in contrast to acute COVID-19, the underlying pathology of *long COVID* remains poorly understood. Symptom burden appears to be independent of the severity of initial infection, with no clear predictors of who will develop ongoing symptoms [10]. Persistent endothelial dysfunction may be responsible for the wide variation in symptoms reported [11,12,13]. There is also some evidence suggesting that altered inflammatory cytokine pathways may persist during convalescence [14].

There has been increasing attention paid to the potential role of vitamin D deficiency in the susceptibility to and the severity of acute COVID-19 [15,16,17]. Vitamin D is a steroid hormone, and deficiency is associated with older age, obesity, ethnicity, chronic medical conditions and geographical location [18]. While the effects of vitamin D deficiency on bone health are well established, the potential pleotropic effects of vitamin D deficiency are becoming increasingly evident. For instance, vitamin D may have an important role to play in severe infections, autoimmune disease and cardiovascular disease [19,20]. Importantly, vitamin D may act as an immunomodulator; this role has gained increasing attention in light of the COVID-19 pandemic [21,22]. Vitamin D levels may also be reduced in the setting of acute inflammation [23]. Furthermore, vitamin D has previously shown to improve symptoms of chronic fatigue and anxiety in patients with these conditions [24,25].

There have been a large number of studies looking at the impact of vitamin D on acute SARS-CoV-2 infection, with variable results reported [26,27,28]. Vitamin D deficiency appears to be associated with severe disease, with some evidence suggesting that vitamin D plays a protective role [29,30]. A recent comprehensive systematic review of 23 studies concluded that while vitamin D deficiency seems to be associated with increased severity and mortality in COVID-19, these findings do not imply causality, and further well-designed prospective studies are required to decipher if deficiency is an epiphenomenon or consequence of the inflammatory response in severe COVID-19 illness [31]. Of note, a recent large mendelian randomization study did not support an association between vitamin D levels and COVID-19 susceptibility, severity or hospitalization in a large consortium of cases and controls [32].

In contrast to the large number of studies investigating the role of vitamin D in acute COVID-19, there are relatively few studies examining the relationship between vitamin D and *long COVID*. This is an important gap in the literature, as the patient population most affected by persistent symptoms following COVID-19 is distinct from that which suffers severe disease in acute infection. *Long COVID* cohorts tend to be young and relatively fit, unlike the older frail patients who develop severe acute disease [33,34]. This suggests that the underlying pathology in *long COVID* is distinct from that seen in acute SARS-CoV-2. Indeed, there have been calls for vitamin D to be measured routinely in all patients attending for post-COVID-19 assessment [35]. Therefore, vitamin D deficiency may be an important modifiable risk factor for persistent symptoms following COVID-19 illness. Low levels of vitamin D have previously been associated with both fatigue and muscle weakness in general populations prior to the COVID-19 pandemic [36,37]. Given that these are the commonest symptoms reported following COVID-19, we hypothesize that persistent fatigue and reduced exercise tolerance following SARS-CoV-2 infection are associated with low levels of vitamin D. In this study, we assess these features using both objective and subjective measures in convalescent COVID-19 patients across the spectrum of initial disease severities and investigate their relationship with convalescent 25-hydroxy vitamin D levels. We also investigate the relationship between vitamin D and circulating pro-inflammatory cytokine C-reactive protein (CRP) and interleukin-6 (IL-6) during convalescence.

## 2. Materials and Methods

### 2.1. Study Setting and Participants

This study was conducted at the post-COVID-19 outpatient clinic at St James’s Hospital, Dublin, Ireland between June and September 2020. This clinic offers appointments to all patients with a positive SARS-CoV-2 polymerase chain reaction (PCR) swab at the institution, including both patients requiring admission during acute infection and those who do not. Assessment of persistent symptoms and phlebotomy for vitamin D measurement were performed on the same day as the clinic assessment. The characteristics of initial infection were recorded, namely the requirement for admission to hospital or the intensive care unit (ICU). Vitamin D supplementation was also recorded. Patients taking vitamin D supplementation were included in the analysis as there was no significant difference in vitamin D levels between supplementation and non-supplementation groups (as detailed in Section 3.1). Body mass index (BMI) was calculated, and a clinical frailty score, operationalized using the Rockwood clinical frailty scale, was assigned to each participant by the clinician at the time of assessment [38].

### 2.2. Laboratory Analysis

Samples for vitamin D analysis included total 25-hydroxy-vitamin D (25(OH)D) (D2 and D3) concentrations, which were quantified by a validated method (Chromsystems Instruments and Chemicals GmbH; MassChrom 25-OH-Vitamin D3/D2) using liquid chromatography-tandem mass spectrometry (LC-MS-MS) (API 4000; AB SCIEX, Framingham, MA, USA) and analysed in the Biochemistry Department of St James’s Hospital (accredited to ISO 15,189). The quality and accuracy of the method was monitored using internal quality controls, participation in the Vitamin D External Quality Assessment Scheme (DEQAS), and the use of the National Institute of Standards and Technology (NIST) 972 vitamin D standard reference material. Vitamin D sufficiency was defined as a serum 25(OH)D concentration ≥50 nmol/L, vitamin D insufficiency as 30 to 49.9 nmol/L, and vitamin D deficiency as <30 nmol/L [39]. IL-6 and CRP were measured in serum by ELISA (Ella ProteinSimple) (normal ranges: IL-6 0–6.2 pg/mL; CRP 0–5 mg/mL).

### 2.3. Symptom Assessment

Fatigue was assessed using the Chalder Fatigue Scale (CFQ-11) [40,41]. Briefly, participants answered eleven questions in relation to physical and psychological fatigue, with reference to the past month in comparison to their pre-COVID-19 baseline. A Likert scale (0–3) measured responses, giving a total score ranging from 0 to 33, with higher scores associated with increasing fatigue [42].

The CFQ-11 also allows differentiation of “cases” versus “non-cases” of fatigue where scores 0 and 1 (Better than usual/No worse than usual) are scored a zero and scores 2 and 3 (Worse than usual/Much worse than usual) are scored a 1 (bimodal scoring). A total score of four or greater was considered to meet the criteria for fatigue. This latter method resembles other fatigue questionnaires [42,43,44,45].

Exercise tolerance and cardiopulmonary and musculoskeletal function were assessed using a six-minute walk test (6MWT) [46,47]. Total distance covered was recorded, while the Modified Borg Scale (MBS) was used to assess perceived exertion during the 6MWT (range 0–10) [48,49].

### 2.4. Statistical Analysis

All statistical analysis was carried out using STATA v15.0 (Stata Statistical Software, College Station, TX, USA), and statistical significance was considered *p* < 0.05. Descriptive statistics are reported as means with standard deviations (SDs) and medians with interquartile ranges (IQRs), as appropriate. Between-group differences were assessed using ANOVA, with post hoc testing as appropriate.

The relationship between vitamin D levels and distance covered for the 6MWT, maximum MBS score, and CFQ score was assessed using linear regression under both unadjusted and adjusted conditions, while the relationship with fatigue case status was assessed using logistic regression. Models were adjusted for age, sex, requirement for admission during initial infection, time from illness to assessment, seasonal vitamin D levels, and use of vitamin D supplementation. Results are reported as β coefficients or odds ratios (ORs) with 95% confidence intervals (CIs) and corresponding *p*-values. Linear models were examined for multi-collinearity by computing variance inflation factors and visually examining residual-versus-fit plots. The relationship between vitamin D status (sufficient, insufficient, deficient) and symptom measures was also assessed in an identical manner.

## 3. Results

### 3.1. Cohort Characteristics

A total of 149 participants (*n* = 88 female, 59.1%) were recruited, with a mean age of 48 years (SD 15). Assessment was performed at a median time from initial infection of 79 days (IQR 67–110). The median vitamin D level in the total cohort was 62 nmol/L (IQR 44–79); *n* = 99 (66%) had vitamin D levels within the sufficient range, *n* = 36 (24%) had insufficient vitamin D levels, and *n* = 14 (9%) were vitamin D deficient. Patients with vitamin D deficiency had significantly shorter time to follow-up compared to those with sufficient levels (*t* = −2.59, *p* = 0.03). There was also a significantly larger proportion of females in the vitamin D sufficient group compared to the deficient group (*t* = −2.77, *p* = 0.02). There were ethnic differences in vitamin D status, with an increased proportion of non-Caucasian patients having vitamin D insufficiency or deficiency. The complete characteristics of the cohort are shown in Table 1.

A variety of vitamin D supplements were taken by *n* = 15 (10%) of participants. Six were taking calcium carbonate/cholecalciferol 500 mg/400 IU combination tablets once per day, with the remaining nine patients taking over-the-counter multivitamins containing cholecalciferol. The median vitamin D level in patients taking supplementation was 65 nmol/L (IQR 45–81), compared with 62 nmol/L (IQR 42–79) in the group without supplementation. The difference between the groups was non-significant (*z* = −0.66, *p* = 0.52).

Fatigue was common in the cohort, with *n* = 86 (57%) meeting the case definition for fatigue, as defined by four or more answers in the more or much more than usual category on the CFQ-11. The median CFS score was 15 (IQR 11–20.5). Similarly, the median MBS score was 3 (IQR 2–5), indicative of moderate exertion. The median distance covered during the 6MWT was 450 m (IQR 390–510). These outcomes were assessed with regard to vitamin D sufficiency, insufficiency or deficiency; there were no differences between vitamin D status and any of the measures of ill health following univariate analysis (Table 1).

### 3.2. Relationship between Post-COVID-19 Ill Health and Vitamin D Levels

We examined the relationship between convalescent vitamin D levels and severity of post-COVID ill health. The relationships were first assessed under unadjusted conditions. There were no associations between vitamin D levels and distance covered during the 6MWT or maximal MBS score during the 6MWT (Figure 1A,B), nor were there any significant differences between vitamin D levels between fatigued cases and non-fatigued cases (Figure 1D). The median vitamin D level for patients meeting the case definition for fatigue was 64 nmol/L (IQR 43–81) compared with 61 nmol/L (IQR 38–75) for non-fatigued cases. There was a significant association between higher vitamin D levels and higher total CFS scores under these unadjusted conditions (Figure 1C).

These relationships were then assessed following adjustment for known confounders of sex, age, requirement for admission during initial infection, season at time of testing, and vitamin D supplementation. We also adjusted for time to follow-up, as this was significantly different on univariate analysis between vitamin D sufficient, insufficient and deficient groups. Following these adjustments there were no significant relationships between vitamin D status and any of the four physical outcomes assessed (Table 2). Female sex was associated with increased fatigue as well as reduced distance for the 6MWT and increased perception of exertion, while increasing age was also associated with reduced distance covered for the 6MWT.

Despite no differences being seen between vitamin D status and physical outcomes following univariate analysis, we examined high-level interactions between these parameters using identical multivariable regression models used previously. This demonstrated no association between vitamin D status and persistent symptoms following SARS-CoV-2 infection, following adjustment for confounders (Table 3). The association between female sex and increasing fatigue was observed again.

Finally, we examined the association between vitamin D levels and circulating CRP and IL-6 during convalescence. The majority of participants had CRP and IL-6 levels within normal limits, and there were no differences between vitamin D status and either convalescent CRP or IL-6 levels (Table 1). There was also no association between convalescent vitamin D levels and CRP (*r*^2^ = −0.08, *p* = 0.35) or IL-6 (*r*^2^ = −0.09, *p* = 0.32) levels.

## 4. Discussion

We assessed the relationship between convalescent vitamin D levels and the cardinal features of *long COVID*, namely reduced exercise tolerance and fatigue. We found that ongoing physical ill-health following SARS-CoV-2 infection is independent of convalescent vitamin D levels following adjustment for confounding variables. Under unadjusted conditions, there was a positive association between fatigue scores and higher vitamin D levels. Our cohort is broadly representative of those reported previously, with a median age of 48 years and female predominance [50]. We included the complete spectrum of acute COVID-19 disease, with just under half the cohort requiring admission to hospital, and 19% requiring ICU admission. This cohort also has a wide range of vitamin D levels recorded, with one-third of patients demonstrating vitamin D insufficiency or deficiency. Vitamin D insufficiency was more common in non-white participants, which has previously been reported in Irish populations [51]. We noted an association between female sex and reduced distance covered for the 6MWT, which is well recognised within healthy populations [52]. The association between female sex and both fatigue and increased perceived exertion following SARS-CoV-2 infection also reflects findings reported elsewhere [53,54]. However, the underlying mechanisms behind the female preponderance for these symptoms remains unclear. The significantly higher proportion of females in the vitamin D–sufficient cohort accounts for the apparent association between fatigue score and higher vitamin D levels under unadjusted conditions. This association is not seen following multivariate analysis.

The lack of association between vitamin D levels and post-COVID-19 fatigue and reduced exercise tolerance is notable. Vitamin D has been associated with chronic fatigue, reduced exercise tolerance, vascular health and affective disorders in pre-pandemic populations [55,56,57]. These are all common symptoms in the aftermath of SARS-CoV-2 infection [58,59,60]. A previous study examined vitamin D levels during initial infection in hospitalised patients and persistent symptoms at eight weeks post illness and also found no relationship [61]. The lack of associations seen emphasise that *long COVID* remains a symptom-driven disease, with no reliable or reproducible biomarker [62,63]. This is also reflected in the levels of CRP and IL-6 recorded, with the majority being within normal range. Furthermore, we see no association with severity of initial infection and subsequent fatigue score or 6MWT performance on multivariate analysis. This is consistent with previous reports demonstrating post-COVID ill health is independent of initial disease severity [64].

We demonstrate that vitamin D levels are not associated with two of the most commonly reported symptoms following COVID-19. However, we have not evaluated its relationship with the myriad of other symptoms occurring in the aftermath of SARS-CoV-2 infection. Notably, two-thirds of our study cohort had vitamin D levels >50 nmol/L. A focused assessment of persistent symptoms in a larger cohort with insufficient or deficient levels of vitamin D would be of value. It is important to note that these blood vitamin D levels represent current status and are not reflective of long-term status (i.e., 3–6 months), which may be important for immune support over a longer time period. Future techniques, such as measuring vitamin D in hair to obtain levels over 6–12 months, may provide additional valuable information [65]. It is likely that these symptoms are multifactorial in aetiology, and as such, further work should be undertaken in this area. The strengths of our study lie in the wide spectrum of initial disease severity included, with both those hospitalised and those managed in the community included, as well as the combination of objective and subjective assessments of symptom burden. We also apply a robust statistical approach to account for potential confounders and reflect an accurate interpretation of the relationship between convalescent vitamin D levels and the persistent symptoms measured.

## 5. Conclusions

We provide an assessment of 149 convalescent SARS-CoV-2 patients at a median of 79 days following infection. We demonstrate a significant symptom burden, with more than half the population meeting the case definition for fatigue. One-third of patients are either vitamin D insufficient or deficient. However, we find no relationship between vitamin D level and persistent symptoms following COVID-19.

## Figures and Tables

**Figure 1 nutrients-13-02430-f001:**
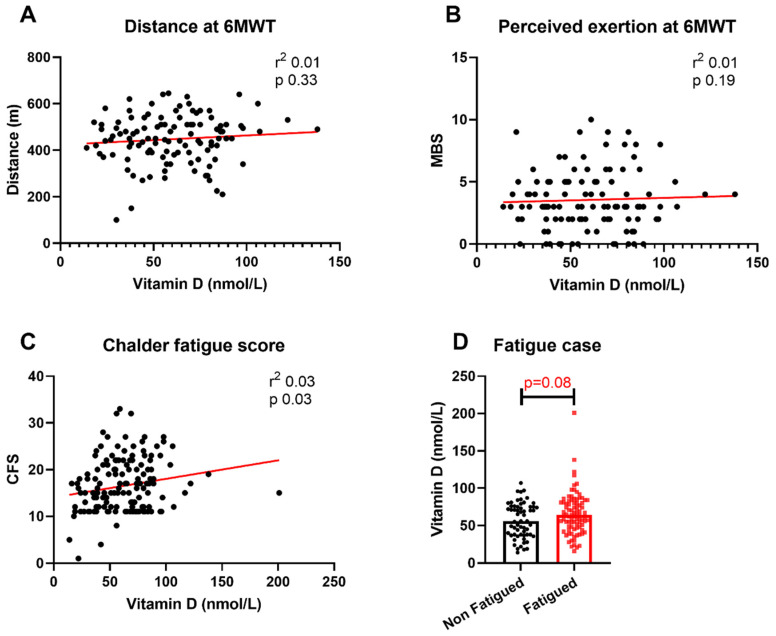
Relationship between vitamin D levels and physical outcomes. Relationship between vitamin D levels and (**A**) distance covered at 6MWT, (**B**) maximal MBS score during 6MWT, (**C**) total CFS score and (**D**) fatigue case status. 6MWT = six-minute walk test; MBS = modified Borg scale; CFS = Chalder Fatigue Score. Differences assessed using unadjusted linear regression and Wilcoxon rank-sum test.

**Table 1 nutrients-13-02430-t001:** Cohort characteristics.

	Total (*n* = 149)	>50 nmol/L (*n* = 99)	30–49 nmol/L	<30 nmol/L	Statistic
			(*n* = 36)	(*n* = 14)	
**Age, years; mean (SD)**	48 (15)	47 (15)	52 (14)	45 (12)	Χ^2^ = 0.79, *p* = 0.12
**Sex, female; n (%)**	88 (59)	66 (66.7)	18 (50)	4 (28.6)	Χ^2^ = 0.26, *p* = 0.01
**Ethnicity; n (%)**					Χ^2^ = 33.1, *p* < 0.001
**-White**	−111 (75)	−81 (82)	−25 (69)	−5 (36)	
**-Asian**	−26 (17)	−15 (15)	−6 (17)	−5 (36)	
**-Hispanic**	−4 (3)	−2 (2)	−1 (3)	−1 (7)	
**-African**	−8 (5)	1 (1)	−4 (11)	−3 (21)	
**Clinical frailty score; median (IQR)**	1 (1–2)	1 (1–2)	2 (1–2)	1.5 (1–2)	Χ^2^ = 1.56, *p* = 0.21
**BMI; median (IQR)**	28 (25–32)	27 (24–33)	29 (27–32)	27 (25–32)	Χ^2^ = 0.79, *p* = 0.68
**Vitamin D; median (IQR)**	62 (44–79)	74 (62–84)	39 (37–45)	22 (19–24)	Χ^2^ = 78.1, *p*<0.001
**CRP, mg/L; median (IQR)**	1 (1–3)	1 (1–3)	1 (1–3)	1 (1–6)	Χ^2^ = 0.23, *p* = 0.08
**IL-6, pg/mL; median (IQR)**	3 (3–4)	3 (3–4)	3 (3–4)	3 (3–6)	Χ^2^ = 0.15, *p* = 0.96
**Vitamin D supplementation; n (%)**	15 (10)	9 (9)	5 (14)	1 (7)	Χ^2^ = 2.42, *p* = 0.67
**Required admission; n (%)**	68 (46)	40 (40)	19 (53)	9 (64)	Χ^2^ = 0.04, *p* = 0.15
**Admitted to ICU; n (%)**	17 (19)	11 (11)	5 (14)	1 (7)	Χ^2^ = 0.93, *p* = 0.67
**Time to OPD, days; median (IQR)**	79 (67–110)	85 (67–112)	75 (67–109)	70 (66–73)	Χ^2^ = 14.97, *p* = 0.04
**Distance at 6MWT, m; median (IQR)**	450 (390–510)	470 (390–510)	443 (390–495)	443 (398–500)	Χ^2^ = 5.29, *p* = 0.46
**Maximal MBS; median (IQR)**	3 (2–5)	3 (2–5)	3 (2–5)	3 (3–4)	Χ^2^ = 2.06, *p* = 0.77
**Fatigue score; median (IQR)**	15 (11–21)	17 (12–22)	15 (11–20)	13 (11–16)	Χ^2^ = 0.65, *p* = 0.01
**Fatigue case; n (%)**	86 (58)	62 (63)	18 (50)	6 (43)	Χ^2^ = 0.13, *p* = 0.21

SD = standard deviation; IQR = interquartile range; ICU = intensive care unit; 6MWT = six-minute walk test; BMI = body mass index; MBS = modified Borg score. Differences assessed using ANOVA with post hoc Tukey test.

**Table 2 nutrients-13-02430-t002:** Multivariable regression analysis of relationship between vitamin D levels and physical health post-COVID-19.

	6MWT Distance		6MWT MBS		CFQ		Fatigue Case	
	β Coefficient (95% CI)	*p* Value	β Coefficient (95% CI)	*p* Value	β Coefficient (95% CI)	*p* Value	Odds Ratio (95% CI)	*p* Value
**Vitamin D**	0.21 (−0.5–1.0)	0.58	−0.01 (−0.1–0.1)	0.78	0.02 (−0.01–0.1)	0.19	1.01 (0.99–1.02)	0.28
**Female sex**	−37.6 (−74–−1.3)	0.04	1.0 (0.02–2.0)	0.04	5.0 (3.1–6.8)	<0.001	4.2 (1.9–9.1)	<0.001
**Age**	−3.5 (−4.9–−2.1)	<0.001	0.02 (−0.1–0.1)	0.28	0.01 (−0.1–0.1)	0.68	1.02 (0.99–1.04)	0.31
**Time to OPD**	0.2 (−0.4–0.8)	0.59	−0.01 (−0.1–0.1)	0.15	−0.02 (−0.1–0.1)	0.86	0.99 (0.98–1.01)	0.70
**Admission**	−30 (−71–11)	0.15	−0.1 (−1.2–1.0)	0.80	0.5 (−1.6–2.6)	0.62	0.9 (0.4–2.2)	0.84
**Season**	7 (−27–41)	0.68	0.6 (−0.3–1.5)	0.22	0.3 (−1.6–2.1)	0.79	1.1 (0.5–2.3)	0.88
**Supplement**	−11 (−75–52)	0.73	0.2 (−1.5–1.9)	0.78	−0.5 (−3.6–2.6)	0.73	1.6 (0.4–5.9)	0.49

Admission = requirement for admission during acute COVID-19; season = season of year at time of vitamin D measurement; supplement = taking vitamin D supplementation; 6MWT = six-minute walk test; MBS = modified Borg scale; CFQ = Chalder fatigue score. Analysis was performed using a single multivariable linear/logistic regression model for each physical outcome.

**Table 3 nutrients-13-02430-t003:** Multivariable regression analysis of relationship between vitamin D sufficiency and physical health post-COVID-19.

	6MWT Distance		6MWT MBS		CFQ		Fatigue Case	
	β Coefficient	*p* Value	β Coefficient	*p* Value	β Coefficient	*p* Value	Odds Ratio	*p* Value
	(95% CI)		(95% CI)		(95% CI)		(95% CI)	
**Vitamin D**								
**-sufficient**	Reference	n/a	Reference	n/a	Reference	n/a	Reference	n/a
**-insufficient**	−4 (−45–36)	0.83	−0.2 (−1.3–0.8)	0.68	−1.0 (−3.1–1.0)	0.33	0.6 (0.3–1.4)	0.23
**-deficient**	−6 (−66–55)	0.85	0.4 (−1.3–2.0)	0.66	−2.7 (−5.9–0.5)	0.09	0.9 (0.2–3.4)	0.89
**Female sex**	−36 (−73–0.7)	0.06	0.9 (0.1–2.0)	0.05	4.7 (2.9–6.6)	<0.001	4.2 (1.9–9.4)	<0.001
**Age**	−3.5 (−4.9–-2.0)	<0.001	0.01 (−0.1–0.1)	0.23	0.01 (−0.1–0.1)	0.75	1.01 (0.9–1.1)	0.24
**Time to OPD**	0.2 (−0.5–0.9)	0.60	−0.01 (−0.1–0.1)	0.18	−0.01 (−0.1–0.1)	0.76	0.99 (0.98–1.01)	0.76
**Admission**	−31 (−72–10)	0.14	−0.2 (−1.3–0.9)	0.78	0.6 (−1.5–2.7)	0.58	0.9 (0.4–2.1)	0.76
**Season**	8 (−26–42)	0.65	0.5 (−0.4–1.4)	0.26	0.4 (−1.5–2.2)	0.69	1.1 (0.5–2.3)	0.82
**Supplement**	−10 (−73–54)	0.77	0.2 (−1.5–1.9)	0.79	−0.5 (−3.6–2.6)	0.75	1.7 (0.5–6.3)	0.44

Admission = requirement for admission during acute COVID-19; season = season of year at time of vitamin D measurement; supplement = taking vitamin D supplementation; 6MWT = six-minute walk test; MBS = modified Borg scale; CFQ = Chalder fatigue score. Analysis was performed using a single multivariable linear/logistic regression model for each physical outcome.

## Data Availability

Data available from the corresponding author upon reasonable request.

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
