# Peer review of "Investigating the Relationship between Vitamin D and Persistent Symptoms Following SARS-CoV-2 Infection"

_nutrients, 2021, doi:10.3390/nu13072430_

Round 1

Reviewer 1 Report

General Comments

The manuscript by Liam Townsend and al. entitled “Investigating the relationship between vitamin D and persistent symptoms following SARS- CoV-2 infection” is dealing with a very interesting and up-to-date problem, such as long lasting symptoms following SARS- Cov-2 infection.

The authors have evaluated whether a relationship was present between Vitamin D levels and fatigue and/or reduced exercise tolerance. The possibility that 25 Oh vitamin D levels might be a biomarkers for long COVID-19 is intriguing, although the population considered is too limited not sufficient to test this hypothesis.

Three well known performance tests have been used: Chalder fatigue scale, six-minutes-walk test and Borg Dyspnea Scale that are appropriate for testing the hypothesis.

A total of 149 convalescent COVID-19 patients were considered in this study and of these, only 86 patients were presenting fatigue symptoms. Less then 50% of the total subjects had a severe form of COVID-19 disease requiring admission to hospital.

The median 25 OH Vitamin D of this total population was 62 nmol/l (about 25 ng/ml), a level considered normal by several scientific societies as well as regulatory agencies. The authors report that only one third of the subjects presented levels of Vitamin D in the insufficiency or deficiency range.

This situation presented in this manuscript is in part resembling a problem often seen in the past 4 years in the extensive studies published in the highest rank journals (J. Internal Medicine, Lancet, New England Journal of Medicine, JAMA). In these studies, either observational or interventional, fractured or diabetic or neoplastic or cardiovascular patients have been considered and no correlation have been found since the average level of vitamin D was normal and less then 20% of the subjects studied were vitamin D deficient. To support this, it must be considered that some positive effects have been reported only in 25 OH vitamin D deficient subpopulation. Moreover, the same authors presenting some negative data, reconsidered the wrong study population, referring, subsequently, to those studies as “research waste” [Bolland et al. BMC, Medical Research Methodology (2018), 18:1010].

In this manuscript  Vitamin D was in the normality range in more then 50% of the subjects considered and this has be better underlined. Vitamin D is an hormone behaving as a supplement, therefore only or mostly deficiency can be related to symptoms.

Of 149 patients, only 86 were presented as fatigue cases and considering the relationship among normal, insufficient and deficient levels of vitamin D, although acceptable from the statistical point of view, the number of subjects with either Vitamin D insufficiency or/and deficiency is very limited. Considering this, some more detailed comments are advisable regarding the relationship, in female, between vitamin D sufficiency and physical health in  table 3.

The authors should rephrase their discussion, and state more strongly that no relationship was present between normal, on average, Vitamin D levels and the 3 tests used for fatigue or reduced tolerance specifically in a small group of long COVID-19 subjects. Moreover what was suggested above, regarding Vitamin D levels and lack of symptoms in conditions prior to COVID-19 Pandemic should be included.

Specific comments

  • Pag 2, “6 lines” form the bottom: COVID, instead of CIVID,
  • In the material and methods section or in the results, it should be stated the vitamin D mean level of 86 fatigue cases.
  • Pag 4. It is stated that 15 subjects were taking vitamin D supplements. What was the Vitamin D level of this subgroup? Since this study is observational, did the data chance if supplemented subjects are excluded. If this subpopulation will not be excluded, the information should be present in the Materials and Methods section.
  • Figure 1 C. A possible explanation, in the discussion, should be offered for significant correlation here presented.
  • Table 2 and Table 3. “Female sex” and significant association between in 6MT MBS, CFQ and Fatigue case should be more extensively commented in the discussion.
  • Less then 50% of the patients required admission to hospital and a small number of them was admitted to ICU. Did this sub-group of more severe patients presented any difference in term of Vitamin D level and fatigue/reduced exercise tolerance? This information could be inserted in the discussion.

Reviewer 2 Report

This manuscript examined the relationship between vitamin D and fatigue and reduced exercise tolerance for 149 patients recruited at a median of 79 days after COVID-19 illness. No relationship between vitamin D and any measure of ongoing ill-health was found following multivariable regression analysis. These results suggest that persistent fatigue and reduced exercise tolerance following COVID-19 are independent of vitamin D.

Comment: “any measure of ongoing ill-health” should be revised to indicate that only a few measures of ongoing ill-health were measured. Not included: loss of smell, hearing, neurological damage, cardiovascular health, other organ damage. In addition, the manuscript should address how persistent symptoms are caused by COVID-19. Most likely, it is due to the cytokine storm, which damages the epithelial layer of many organs including the vascular system. Thus, higher vitamin D status during COVID-19 should result in reduced risk of severity of persistent symptoms. Some of these publications as well as additional ones the authors should search for should be discussed.

COVID-19-Associated Cardiovascular Complications.

Lee CCE, Ali K, Connell D, Mordi IR, George J, Lang EM, Lang CC.Diseases. 2021 Jun 29;9(3):47. doi: 10.3390/diseases9030047.

The Neurological Manifestations of Post-Acute Sequelae of SARS-CoV-2 infection.

Moghimi N, Di Napoli M, Biller J, Siegler JE, Shekhar R, McCullough LD, Harkins MS, Hong E, Alaouieh DA, Mansueto G, Divani AA.Curr Neurol Neurosci Rep. 2021 Jun 28;21(9):44. doi: 10.1007/s11910-021-01130-1.

Cardiac sequelae after COVID-19 recovery: a systematic review.

Ramadan MS, Bertolino L, Zampino R, Durante-Mangoni E; Monaldi Hospital Cardiovascular Infection Study Group(∗).Clin Microbiol Infect. 2021 Jun 22:S1198-743X(21)00335-9. doi: 10.1016/j.cmi.2021.06.015

Global prevalence of prolonged gastrointestinal symptoms in COVID-19 survivors and potential pathogenesis: A systematic review and meta-analysis.

Yusuf F, Fahriani M, Mamada SS, Frediansyah A, Abubakar A, Maghfirah D, Fajar JK, Maliga HA, Ilmawan M, Emran TB, Ophinni Y, Innayah MR, Masyeni S, Ghouth ASB, Yusuf H, Dhama K, Nainu F, Harapan H.F1000Res. 2021 Apr 19;10:301. doi: 10.12688/f1000research.52216.1.

Medical sequels of COVID-19.

Peramo-Álvarez FP, López-Zúñiga MÁ, López-Ruz MÁ.Med Clin (Barc). 2021 May 27:S0025-7753(21)00289-X. doi: 10.1016/j.medcli.2021.04.023. 

Understanding Viral Infection Mechanisms and Patient Symptoms for the Development of COVID-19 Therapeutics.

Choi HM, Moon SY, Yang HI, Kim KS.Int J Mol Sci. 2021 Feb 9;22(4):1737. doi: 10.3390/ijms22041737.

Harnessing the immune system to overcome cytokine storm and reduce viral load in COVID-19: a review of the phases of illness and therapeutic agents.

Khadke S, Ahmed N, Ahmed N, Ratts R, Raju S, Gallogly M, de Lima M, Sohail MR.Virol J. 2020 Oct 15;17(1):154. doi: 10.1186/s12985-020-01415-w.

Murine-beta-coronavirus-induced neuropathogenesis sheds light on CNS pathobiology of SARS-CoV2.

Chakravarty D, Das Sarma J.J Neurovirol. 2021 Apr;27(2):197-216. doi: 10.1007/s13365-021-00945-5.

Coronavirus disease 2019 and cardiovascular complications: focused clinical review.

Saeed S, Tadic M, Larsen TH, Grassi G, Mancia G.J Hypertens. 2021 Jul 1;39(7):1282-1292. doi: 10.1097/HJH.0000000000002819.

Vitamin D may protect against multiple organ damage caused by COVID-19.

Aygun H.Bratisl Lek Listy. 2020;121(12):870-877. doi: 10.4149/BLL_2020_143.

Putative roles of vitamin D in modulating immune response and immunopathology associated with COVID-19.

Kumar R, Rathi H, Haq A, Wimalawansa SJ, Sharma A.Virus Res. 2021 Jan 15;292:198235. doi: 10.1016/j.virusres.2020.198235. 

Vitamin D deficiency as a predictor of poor prognosis in patients with acute respiratory failure due to COVID-19.

Carpagnano GE, Di Lecce V, Quaranta VN, Zito A, Buonamico E, Capozza E, Palumbo A, Di Gioia G, Valerio VN, Resta O.J Endocrinol Invest. 2021 Apr;44(4):765-771. doi: 10.1007/s40618-020-01370-x.

Cytokine Storm in COVID-19: "When You Come Out of the Storm, You Won't Be the Same Person Who Walked in".

Castelli V, Cimini A, Ferri C.Front Immunol. 2020 Sep 2;11:2132. doi: 10.3389/fimmu.2020.02132. 

Unraveling the mystery of Covid-19 cytokine storm: From skin to organ systems.

Garg S, Garg M, Prabhakar N, Malhotra P, Agarwal R.Dermatol Ther. 2020 Nov;33(6):e13859. doi: 10.1111/dth.13859. Epub 2020 Jul 7.PMID: 32559324 Free PMC article. Review.

The Conundrum of 'Long-COVID-19': A Narrative Review.

Garg M, Maralakunte M, Garg S, Dhooria S, Sehgal I, Bhalla AS, Vijayvergiya R, Grover S, Bhatia V, Jagia P, Bhalla A, Suri V, Goyal M, Agarwal R, Puri GD, Sandhu MS.Int J Gen Med. 2021 Jun 14;14:2491-2506. doi: 10.2147/IJGM.S316708. 

Vitamin D deficiency is associated with COVID-19 positivity and severity of the disease.

Demir M, Demir F, Aygun H.J Med Virol. 2021 May;93(5):2992-2999. doi: 10.1002/jmv.26832. 

On the other hand, “A previous study examined the vitamin D levels during initial infection in
hospitalised patients and persistent symptoms at 8 weeks post illness and also found no
relationship (50)” as mentioned in the manuscript.

Of note, a recent large mendelian randomization study did not
support an association between vitamin D levels and COVID-19 susceptibility, severity or
hospitalization in a large consortium of cases and controls (23).

Comment: As noted in Ref. 23:

“Our study still has limitations. First, our results do not apply to individuals with vitamin D deficiency, and it remains possible that truly deficient patients may benefit from supplementation for COVID-19-related protection and outcomes.”

That result was not surprising since MR studies seldom confirm findings from observational studies or clinical trials regarding vitamin D. Another problem is that it does not include data related to vitamin D supplementation, so those who supplement have much higher 25OHD concentrations than those who do not and better health outcomes.

“A recent comprehensive systematic review of 23 studies concluded that whilst vitamin D deficiency seems to
be associated with increased severity and mortality in COVID-19, these findings do not
imply causality and further well-designed prospective studies are required to decipher if
deficiency is an epiphenomenon or consequence of the inflammatory response in severe
COVID-19 illness (22).”
See:
Effects of a 2-Week 5000 IU versus 1000 IU Vitamin D3 Supplementation on Recovery of Symptoms in Patients with Mild to Moderate Covid-19: A Randomized Clinical Trial.

Sabico S, Enani MA, Sheshah E, Aljohani NJ, Aldisi DA, Alotaibi NH, Alshingetti N, Alomar SY, Alnaami AM, Amer OE, Hussain SD, Al-Daghri NM.Nutrients. 2021 Jun 24;13(7):2170. doi: 10.3390/nu13072170.

All statistical analysis was carried out using STATA v15.0 (Texas, USA)

Please include city names for companies.

Significant digits. The general rule is that no more non-zero digits should be given than are justified by the uncertainty of the value.

See "Too many digits: the presentation of numerical data"

https://www.ncbi.nlm.nih.gov/pmc/articles/PMC4483789/

If the uncertainty is greater than about 7%, only two non-zero digits are justified.

P values should be given to two decimal places unless the first two are 00 or the number lies between 0.045 and 0.050.

Thus

Age, years; mean (SD) 47.6 (14.7) 46.5 (15.0) 52.0 (14.3) 44.7 (12.3) Χ2=0.79, p=0.68
Sex, female; n (%) 88 (59.1) 66 (66.7) 18 (50) 4 (28.6) Χ2=0.26, p=0.88

Should be

Age, years; mean (SD) 48 (15) 47 (15) 52 (14) 45 (12) Χ2=0.79, p=0.68
Sex, female; n (%) 88 (59) 66 (67) 18 (50) 4 (29) Χ2=0.26, p=0.88

Percentages should be given in whole numbers due to the small N.

the r2 values in Figure 1 should be revised. That can be done by reducing the number of digits to 4 in the graphical program.

Please review all numbers in abstract, text, tables, and figures and adjust accordingly

Round 2

Reviewer 1 Report

The manuscript by Liam Townsend and al. entitled “Investigating the relationship between vitamin D and persistent symptoms following SARS- CoV-2 infection” is dealing with long lasting symptoms following SARS- Cov-2 infection.

The manuscript has been revised by the authors in a satisfactory fashion.

It is important that the authors added data and comments regarding IL-6 and CPR levels in these patients. It is appreciable that the authors added a comment regarding the basal levels of vitamin D as real importance factor when considering any possible effect of its deficiency.

2.1 “Patients taking vitamin D supplementation were included in the analysis as there was no significant difference in vitamin D levels between supplementation and non-supplementation groups”. For a better comprehension by the readers, It should be anticipated here where this important issue is presented in the results/able section

Author Response

Many thanks for the comments regarding the changes made. 

We have now included a reference to results section 3.1 in the methods to highlight the lack of significant difference between vitamin D levels in patients with and without supplementation. 

Reviewer 2 Report

Thanks for making the suggested revisions.

I now recommend the manuscript for publication.

Author Response

We thank the reviewer for their comments regarding the changes made.